# Genome Evolution and the Future of Phylogenomics of Non-Avian Reptiles

**DOI:** 10.3390/ani13030471

**Published:** 2023-01-29

**Authors:** Daren C. Card, W. Bryan Jennings, Scott V. Edwards

**Affiliations:** 1Department of Organismic & Evolutionary Biology, Harvard University, Cambridge, MA 02138, USA; 2Museum of Comparative Zoology, Harvard University, Cambridge, MA 02138, USA; 3Department of Evolution, Ecology & Organismal Biology, University of California Riverside, Riverside, CA 92521, USA; 4Departamento de Ecologia, Programa de Pós-Graduação em Ecologia e Evolução, Universidade do Estado do Rio de Janeiro, Rio de Janeiro 20550-013, Brazil; 5Departamento de Vertebrados, Museu Nacional, Universidade Federal do Rio de Janeiro, Rio de Janeiro 20490-040, Brazil

**Keywords:** anonymous loci, GC content, genome size, isochores, karyotype, natural history, reduced representation, repetitive elements, sex determination and chromosomes, target capture, ultraconserved elements

## Abstract

**Simple Summary:**

As a group of organisms, non-avian reptiles, most of which are the ~11,000 species of lizards and snakes, are an extraordinarily diverse group, displaying a greater diversity of genetic, genomic, and phenotypic traits than mammals or birds. Yet the number of genomes available for non-avian reptiles lags behind that for other major vertebrate groups. Here we review the diversity of genome structures and reproductive and genetic traits of non-avian reptiles and discuss how this diversity can fuel the next generation of whole-genome phylogenomic analyses. Whereas most higher-level phylogenies of non-avian reptile groups have been driven by a group of markers known as ultraconserved elements (UCEs), many other types of markers, some with likely greater information content than UCEs, exist and are easily mined bioinformatically from whole-genomes. We review methods for bioinformatically harvesting diverse marker sets from whole genomes and urge the community of herpetologists to band together to begin collaboratively constructing a large-scale, whole-genome tree of life for reptiles, a process that has already begun for birds and mammals. Such a resource would provide a much-needed high-level view of the phylogenetic relationships and patterns of genome evolution in this most diverse clade of amniotes.

**Abstract:**

Non-avian reptiles comprise a large proportion of amniote vertebrate diversity, with squamate reptiles—lizards and snakes—recently overtaking birds as the most species-rich tetrapod radiation. Despite displaying an extraordinary diversity of phenotypic and genomic traits, genomic resources in non-avian reptiles have accumulated more slowly than they have in mammals and birds, the remaining amniotes. Here we review the remarkable natural history of non-avian reptiles, with a focus on the physical traits, genomic characteristics, and sequence compositional patterns that comprise key axes of variation across amniotes. We argue that the high evolutionary diversity of non-avian reptiles can fuel a new generation of whole-genome phylogenomic analyses. A survey of phylogenetic investigations in non-avian reptiles shows that sequence capture-based approaches are the most commonly used, with studies of markers known as ultraconserved elements (UCEs) especially well represented. However, many other types of markers exist and are increasingly being mined from genome assemblies in silico, including some with greater information potential than UCEs for certain investigations. We discuss the importance of high-quality genomic resources and methods for bioinformatically extracting a range of marker sets from genome assemblies. Finally, we encourage herpetologists working in genomics, genetics, evolutionary biology, and other fields to work collectively towards building genomic resources for non-avian reptiles, especially squamates, that rival those already in place for mammals and birds. Overall, the development of this cross-amniote phylogenomic tree of life will contribute to illuminate interesting dimensions of biodiversity across non-avian reptiles and broader amniotes.

## 1. Introduction

Amniote vertebrates are an important clade encompassing humans, model organisms such as mouse and chicken, and many other non-model taxa, which has collectively become the most well-studied radiation of eukaryotes [1,2]. Among amniotes, there are two major evolutionary lineages—mammals and reptiles—that vary in several major natural history characteristics, such as the presence of hair versus scales, the production of milk for nourishing young, and the features of the skeletal system, especially skull structure and jaw articulation. Significant variation also exists among reptiles, resulting in four major groups that are often studied in isolation from mammals and one another: (1) birds (Class Aves), (2) crocodylians (Class Reptilia, Order Crocodylia), (3) turtles (Class Reptilia, Order Testudines), and (4) squamate reptiles (Class Reptilia, Order Squamata) [3]. Dinosaurs (including birds), crocodylians, and turtles form one major clade of reptiles, Archosauromorpha, whereas squamates and the unique taxon tuatara (Class Reptilia, Order Sphenodontia, *Sphenodon punctatus*) form the other major reptilian clade, Lepidosauromorpha. Archosauromorpha and Lepidosauromorpha diverged approximately 281 million years ago (MYA) and most of the major reptilian lineages had emerged by approximately 250 MYA [4,5]. Based on numbers of extant species, there are large differences in the diversity of each major reptilian lineage (Figure 1). Tuatara (1 species), crocodylians (27 species) and turtles (356 species) have relatively few species [6,7] whereas mammals, birds, and squamates comprise the vast majority of amniotes. In contrast to mammals and birds, whose species counts have been relatively stable (current counts of 6495 [https://www.mammaldiversity.org/ (accessed on 1 December 2022)] and 10,906 species [https://birdsoftheworld.org/bow/home (accessed on 1 December 2022)], respectively), new squamates continue to be described at a high rate, resulting in thousands of new species having been recognized in the last 10 years and a total species count (11,349 species of squamates as of March 2022; [6,7]) that now surpasses birds, which had long been regarded as the most species-rich group of tetrapods (Figure 1).

The recent rapid taxonomic growth of non-avian reptiles, especially squamates, has paralleled early growth and development of genomic resources in these clades, which is beginning to enable a range of investigations in the established but rapidly evolving field of phylogenomics. Phylogenomics is the field of study concerned with using genome-wide data to infer the evolution of genes, genomes, and the tree of life [33]. Phylogenomics datasets are a product of complex patterns of evolution evident across genomic loci, many of which are influenced by various natural history characteristics of the focal taxa, and the imperfect process of producing and extracting meaningful information from genomics data. Therefore, phylogenomics investigations are both motivated and confounded by the natural histories of the taxa of interest and the underlying characteristics of the genomics data [34,35,36]. Moreover, the importance of reference genomes in phylogenomic investigations is growing, and interest is also increasing in using phylogenomics approaches to study the evolutionary history and unique natural histories of non-avian reptiles [15,37]. In anticipation of these developments, here, we review and discuss the rich natural histories and available reference genomes of non-avian reptiles and considerations for future phylogenomics investigations based on genomic resources in these lineages.

## 2. Non-Avian Reptiles Are Highly Variable in Physical Traits with Strong Links to the Genome

Non-avian reptiles—turtles, crocodylians, squamates, and tuatara—exhibit many interesting natural history characteristics ranging from physical traits to the composition and structure of the genome [32]. Physical traits that are normally invariant in well-studied amniotes such as mammals and birds are often variable across non-avian reptiles and even within certain reptile clades, making these lineages interesting and important for many biological investigations. Differences in sex determination are evident among non-avian reptiles and two major forms of sex determination have evolved in amniotes: (1) genetic sex determination (GSD), in which biological sex is determined genetically by the presence, absence, or dosage of a particular locus or allele during development, and (2) environmental sex determination (ESD), in which environmental conditions during development, often temperature, controls sex, normally resulting in clutches that are largely or exclusively one sex due to incubation conditions [8]. Whereas mammals and birds are well known examples of clades with only GSD, crocodylians and the tuatara are clades where ESD apparently functions exclusively (Figure 1; see caption for the details of the datasets and their summarization) [38,39,40]. In contrast to this pattern, turtles and squamates each are characterized by species or clades with either GSD or ESD (Figure 1) [8,41,42,43,44] and some interesting examples in which environmental temperature can override known GSD [45,46,47,48,49,50,51]. Overall, squamate reptile sex determination remains poorly understood relative to other amniote clades due to the complexity of sex determination and large numbers of apparent transitions between sex determination mechanisms across squamate species studied so far, although such patterns in squamates also offer an unparalleled opportunity to understand the genetics and evolution of all forms of amniote sex determination.

The complex interplay between environment and organism development that characterizes aspects of sex determination in non-avian reptiles also functions to drive interesting and complex patterns in the evolution of reproductive mode in these lineages. As is the case for most amniotes, sexual reproduction dominates among non-avian reptiles, and like birds and unlike most mammals, all turtles, crocodylians, and the tuatara are oviparous (Figure 1) [10,52,53]. Squamate reptiles, on the other hand, exhibit all major modes of sexual reproduction known from amniotes—oviparity, viviparity, and oviviviparity—and also reproduce asexually via various modes of parthenogenesis (Figure 1) [11,53,54]. Sexual reproductive mode can turnover rapidly in squamate reptiles [55] and numerous squamate species are capable of reproduction via both oviparity and viviparity (e.g., *Zootoca vivipara*, *Lerista bougainvillii*, and *Saiphos equalis*; see [56,57,58,59,60,61]), a situation that has driven the hypothesis that uterine retention is selectively advantageous in cooler environments [62,63]. Squamates can also reproduce via obligate parthenogenesis (Figure 1), resulting in species or populations composed entirely of females, including in certain geckos (*Lepidodactylus lugubris* [64] and *Hemidactylus garnotii* [65]), the well-known ‘flowerpot snake’ (*Indotyphlops braminus* [66]), and several hybrid species from the genera *Cnemidophorus*/*Aspidoscelis* [67] and *Darevskia* [68]. Numerous examples of facultative parthenogenesis have recently been documented in captive squamates, including the Komodo dragon [69] and various snakes [70,71,72,73,74,75,76], and wild populations of pit vipers [77]. As was the case with sex determination, squamate reptiles are an ideal group for investigating the genetics and evolution of reproductive mode and unappreciated examples of unique reproductive modes likely remain to be discovered. Overall, non-avian reptiles possess unparalleled variation in two major natural history traits, sex determination and reproductive mode, which each drive complex evolutionary patterns genome-wide.

## 3. Substantial Variation in Genome Size and Karyotype among Non-Avian Reptiles

At the cellular level, genome size and karyotype comprise an important aspect of biology that can impact other aspects of natural history, including physical traits and patterns of genetic variation [44]. Genome size, in particular, has strong links with the activity of repetitive elements, organism longevity, metabolic rate, and the rate of development [78,79], and has a known impact on cellular physiology, nuclear volume, and overall cell size [80]. Genome size can be measured by mass or by the combined length of all chromosomes and these measures generally correspond 1:1, such that 1 picogram (pg) of DNA corresponds to a 1 gigabasepair (Gbp) genome. Genome size varies greatly among amniotes, generally ranging from a mean of 1.4 pg in birds to 3.2 pg in mammals, and the range of genome sizes in reptiles alone is similarly broad (Figure 1). Both turtles and crocodylians have mean genome sizes that are similar to mammals at 2.8 and 3.0 pg, respectively, and the genome of the tuatara is the largest of any amniote studied to date at 5 pg (Figure 1). Squamate genome sizes are more tightly distributed around an intermediate genome size between birds and mammals at 2.1 pg (Figure 1). Previous investigations of the evolution of genome size in reptiles has yielded nuanced conclusions about the rates of genome size evolution, which have been inferred to be gradual overall [81], but potentially faster in taxa with larger genomes [82].

Karyotypic variation in reptiles is also high relative to what is observed in mammals due mainly to the presence of microchromosomes in several reptile clades [83]. Microchromosomes are approximately half the size of macrochromosomes on average [84] and have higher GC content [85], gene densities [86], and recombination rates [87] and lower densities of repetitive elements [85]. Recent studies of microchromosomes, first in snakes [37] and since more broadly [88,89], indicate that they may have unique functional characteristics relative to macrochromosomes, such as higher rates of interchromosomal contacts between loci of chromosomes in the nucleus of cells. Aside from birds, microchromosomes are present in squamates, tuatara, and turtles, but absent in crocodylians (Figure 1) [90,91,92]. Mammal and bird karyotypes have been particularly well-studied but karyotypical variation for all amniote lineages is well known [32,44]. A mean haploid chromosome count ranging from approximately 17 to 20 describes most lineages of amniotes, including Crocodylia, Rhyncocephalia, Squamata, and Mammalia, although Mammalia has a far greater variance in haploid chromosome count than the other lineages (Figure 1). Birds have a similarly broad distribution in haploid chromosome count but a substantially larger number of chromosomes on average (36.2), whereas turtles have an intermediate mean haploid chromosome count of 25.5 (Figure 1). The breadth of karyotypic diversity in reptiles far exceeds what is observed in mammals, especially when including birds, making this clade ideal for investigating karyotype evolution in amniotes.

The mechanism of sex determination impacts the evolution of sex chromosome systems in the case of GSD, which can result in homomorphic sex chromosomes—sex chromosomes that are superficially similar and hard to identify as linked to sex—and two forms of heteromorphic sex chromosomes—sex chromosomes that are able to be distinguished in males or females due to the evolution of a degenerated chromosome. Heteromorphic chromosomes were originally observed using cytogenetic methodologies and largely continue to be identified in this manner, although detecting less obvious homomorphic chromosomes is also possible, but has been rarely pursued due to increased difficulty. In heteromorphic sex chromosome systems, the evolution of a visually degenerated chromosome can result from sexual conflict, which is overcome through the suppression of recombination via inversions. The degenerated sex chromosome can be inherited paternally, resulting in a XY sex chromosome system, or maternally, resulting in a ZW sex chromosome system [93,94,95]. Again, whereas mammals and birds are all characterized by a common sex chromosome system (XY and ZW, respectively), non-avian reptiles have far more nuanced variation in the form of sex chromosomes across different lineages. In the tuatara and crocodylians, ESD putatively results in lower genetic sexual conflict, resulting in (visually) homomorphic sex chromosomes [96,97,98]. Homomorphic sex chromosomes have also been observed in turtles and squamates, especially those species where ESD is known, but both of these lineages also exhibit species with XY and ZW sex chromosomes [43,44]. The greatest known variation in sex determination and sex chromosome systems is evident in geckos [99]. However, our knowledge of sex determination/chromosomes is still relatively incomplete in non-avian reptiles, especially squamates, and new, interesting phylogenetic patterns of sex chromosomes are regularly being discovered, including the recent discovery of largely homomorphic XY sex chromosomes that evolved independently in Henophidian snakes [100] that violated long-held assumptions that all snakes possessed ZW sex chromosomes [101,102]. Altogether, non-avian reptiles possess the most complex evolutionary patterns of sex chromosome systems of all amniotes, with squamates emerging as a relatively powerful system for interrogating the evolution of sex chromosomes. Finally, we focus here on phylogenomics using the nuclear genome, and do not discuss mitogenomics. However, we note that non-avian reptiles have interesting patterns of evolution of the mitochondrion that should be considered in phylogenomic investigations (e.g., see [103,104]), such as a snake-specific duplicate control region and high rates of adaptive evolution of snake mitochondrial metabolic proteins [105,106].

## 4. Dynamic Features of Sequence Composition in Non-Avian Reptiles

At the sequence level, non-avian reptile genomes are characterized by several unique characteristics that may impact downstream phylogenomics investigations. Non-avian reptile genomes contain a diverse repertoire of repetitive elements that is only beginning to be explored. Most knowledge of amniote repeat element landscapes is based on early genomic investigations in mammals and birds [85,107,108,109], where there is a dichotomy in genomic architecture. Mammals have a relatively rich diversity of repeat elements that form a substantial portion of the genomes of these organisms (e.g., at least 50% of the human genome [107], with other mammals having similar patterns), correlating with larger genomes. In contrast, bird genomes are generally relatively streamlined, containing much less repeat diversity dominated largely by chicken repeat (CR1) long interspersed nuclear elements (LINEs) that form a much smaller portion of the already much smaller genomes of most bird species (typically < 20% of ~1 Gbp genomes; [85,110,111]). Early studies of non-avian reptile repeat landscapes using BAC-end sequences revealed high diversity in tuatara and various squamate lineages [112,113]. More recent studies based on whole-genome sequences have only emerged since 2011 and have established an additional dichotomy in the evolution of repeat landscapes between the non-avian Archosauromorpha (crocodylians and turtles) and Lepidosauromorpha (squamates and tuatara). Crocodylians and turtles have relatively homogeneous repeat landscapes that superficially resemble birds and a reduced rate of new TE family invasion/evolution [16], although larger proportions of the genomes of these species are comprised of repeat elements (>35% [16,17]). In contrast, squamates and tuatara have an extremely rich diversity of repeat elements that exceeds the diversity of mammals. As could be hypothesized given the unique evolutionary history of the tuatara, a large proportion of the genome of this species consists of repeats, and this diverse repeat landscape is unique from any other amniote [18]. Squamates, despite showing a fairly even distribution of genome sizes, have large amounts of variation in the proportions of their genomes composed of repetitive elements, ranging from lower proportions in most lizards of ~30–40% to higher proportions in many colubroid snakes of 50% or greater. Squamate genome repeat landscapes are dominated by three types of LINE families (CR1, BovB, and L2) and have high proportions of DNA transposons, in contrast with other amniote genomes where one LINE family typically dominates [15,114,115]. Moreover, while other amniotes have fairly inactive repeat landscapes, where only one or a few repeats has continued to proliferate in the genomes of these organisms (e.g., L1 LINEs or Alu elements in humans [107]), several repeat types, subtypes, and families appear simultaneously active in squamate genomes [15]. Indeed, the patterns of repeat evolution observed in squamates [15] challenge the accordion model of repeat element evolution in vertebrate genomes that was based on data from mammals and birds [19]. Moreover, the striking phylogenetic pattern of higher repeat element proportions characterizing the genomes of snakes and especially the venomous lineages of snakes has led to speculation that the seeding of repetitive loci, especially microsatellites, which may drive the rapid evolution of tandemly duplicated gene families that are important in evolutionary novelties in these clades: *Hox* genes and the serpentine body plan and various toxin gene families that function in venom [116,117,118]. Finally, although most repeat element activity is limited to the nucleus, there are many documented cases of horizontal transfer of repetitive elements between divergent lineages, including non-avian reptiles, apparently mediated by viruses or blood-sucking ectoparasites [15,114,119,120,121,122,123,124,125,126,127,128].

Even at the level of the nucleotide, studies of GC content indicate non-avian reptile genomes have unique features. Overall GC content varies greatly across amniotes. Mammals and birds have similar GC content with a mean across species of ~41–42%, although mammalian genomes have greater variation in GC content (Figure 1). Squamate reptile genomes have a similar mean GC content, but far greater variation than even mammalian genomes, whereas turtle and crocodylian genomes have elevated GC content (43.9% and 43.8%, respectively) and the tuatara has by far the highest GC content known from any amniote (47%; Figure 1). The composition of bases across the genome is not uniform and large genomic tracts (>100 kb) with relatively homogenous, biased base composition—often called isochores—can form. Isochores are well defined in mammalian and avian genomes but generally absent in fish and amphibians [129,130]. GC-rich isochores, in particular, correlate with several other genomic features, such as recombination rate [131], gene density [132], epigenetic modifications [133], intron length [134], and replication timing [135]. The correlation between GC content and recombination rate is particularly profound for genome biology and evolution, especially in light of documented GC-based repair biases, and mechanistic links between GC content and recombination rate may be a strong driver of variation in recombination rate across the genome [136,137,138,139,140,141].

Early investigations of isochores in reptiles first used CsCl fractionation [142] and, later, GC values at third-codon positions as a proxy for isochores (GC3 [23,24,143,144])—a practice that has since come into question because GC3 only explains a small proportion of variation in the GC content in the regions flanking genes [145]—although ideally such investigations are based on high-quality genome assemblies (e.g., [25,26,146]). Analysis of the first non-avian reptile genome, that of the green anole (*Anolis carolinensis* [147]), established that there was little evidence for isochore structure in this species relative to what is observed in mammals and chicken [27]. However, a subsequent investigation questioned this result [28] and an additional study established that snake genomes have a higher degree of GC-isochore structure than seen in *Anolis* [115], indicating more complex evolutionary patterns of isochore structure in squamate reptiles that remain to be thoroughly investigated. Reference genomes suggest that turtles and crocodylians also show patterns consistent with isochore structure [29,30]. Despite retaining (or perhaps secondarily evolving) isochores, snakes have lower GC content than *Anolis* and the evolution of GC at third codon positions (GC3) trends towards AT richness, in contrast to the GC bias notable in mammalian genomes [115]. Beyond GC content, investigations of nucleotide substitution patterns indicates that squamates generally have higher substitution rates that are similar to those from mammals [16,115], with interesting bursts in the rates of evolution associated with root branches of snakes and colubroid snakes [115]. In contrast, birds have modest substitution rates and analyses of turtle and crocodylian genomes show these lineages have extremely slow substitution rates [16]. Overall, evolution in the composition of non-avian reptile genomes has resulted in remarkably different genomic environments in these lineages, which will need to be taken into account during downstream phylogenomics investigations.

## 5. Summary of Available Reference Genomes for Non-Avian Reptiles

Despite possessing a range of interesting natural histories, genomics resources, and therefore phylogenomics investigations, in non-avian reptiles have only emerged since the publication of the first non-avian reptile genome in 2011, that of the green anole [147]. Since this release, genomic resources for increasing numbers of non-avian reptiles have emerged. These genomic resources are most often available from National Center for Biotechnology Information (NCBI), but other data repositories can be used (e.g., DNAZoo and GenomeArk) and a growing number of sequencing initiatives are targeting non-avian reptiles for reference genomes, making it difficult to collate all resources available for these clades. While genomes are available from relatively large proportions of smaller reptilian clades (i.e., crocodylians, turtles, and tuatara), genomic resources in the species-rich squamate reptiles have been slower in developing than similar resources in mammals and birds, where reference genomes have been constructed for approximately 9% and 6% of species, respectively (Figure 2). However, in recent years the pace of sequencing has increased in non-avian reptiles, especially squamates, due to technological advances and improving economics. An exhaustive accounting indicates there are 165 publicly available and 23 announced (i.e., expected in the future) non-avian reptile reference genomes (Figure 3; see caption for the details of the datasets and their summarization). These genomes collectively represent 139 reptilian species, with redundancy in the form of multiple assemblies of varying quality from the same source material and multiple assemblies sourced from different animals, sometimes representing distinct populations (Figure 3). Reference genomes are available for 31 (9% of known species) turtle species, 4 (15%) crocodylian species, 1 (100%) rhyncocephalian, and 84 (<1%) squamate species. Most non-avian reptile genomes have been released since 2020 (Figure 2 and Figure 3). Moreover, there are significant differences in assembly characteristics (i.e., length and GC content) and quality (i.e., N50s and BUSCO scores) between genomes due to technical differences in assembly production and evolutionary differences in the genomic characteristics of amniotes (Figure 3).

## 6. Why Are There So Few Genomes of Non-Avian Reptiles?

As we have seen, genome sequencing in non-avian reptiles has lagged behind progress in birds, where there are now hundreds of genomes and an increasing number based on long-read sequencing [148]. Additionally, there is a paucity of long-read, high-quality or even chromosome-scale genomes from non-avian reptiles. This paucity likely stems from the academic orientation of the many biologists interested in herpetology. Few if any reptiles can claim the exalted status of a ‘model organism’, especially in the fields of genetics, developmental biology, or cell biology. This is not to say that non-avian reptiles cannot serve as important models for many fields, such as ecology and adaptive radiation: the large number of studies on *Anolis* lizards is clear evidence of this [164]. Nevertheless, the fields for which non-avian reptiles are models tend not as yet to be the fields that require genomes. Of course, there have been several studies that have effectively linked genome variation and ecology in non-avian reptiles, especially for *Anolis* [165,166]. In many ways, the availability of a high-quality genome from *Anolis carolinensis* in 2011 has attracted investigators to that species in diverse contexts; however, by the metrics we use in Figure 3, such activity would not increase the number of non-avian reptile genomes. The increased number of genomes from the genus *Anolis* is beginning to reveal the potential of comparative genomics in non-avian reptiles [167,168,169]. However, as has been evident in ornithology, it takes a strong, ambitious, and sustained focus by the research community on comparative biology, as well as the availability of multiple models for other fields, such as molecular and cellular biology, to drive the accumulation of genomes from multiple species. For example, birds are models in neuroscience (zebra finch, *Taeniopygia guttata*) and developmental biology (chicken, *Gallus gallus*) that are useful enough to compete for priority for particular scientific problems with well-funded models of genetics and genomics, such as mice.

In the era before whole-genome sequencing became routine, several researchers put in place important genomic resources for non-avian reptiles, such as BAC libraries, cell lines, and short-read sequence archives that helped move the field forward [170,171,172]. Several BAC libraries, such as those for the tuatara (*Sphenodon punctatus*) and western painted turtle (*Chrysemys picta*) helped fuel subsequent genome projects of these species, or refinement of assemblies via mapping [18,171,173], and provided useful resources for early phylogenomic analyses [174]. Additionally, some of the first glimpses of the structure of non-avian reptile genomes—including larger-scale observations of GC content [113,175], transposable element abundances [81,112], and non-coding conserved elements [172,176]—came from such resources. Many of these early observations of genome structure in non-avian reptiles were only indirect, and have been vastly improved upon with better tools and direct genome-scale analyses [15,37,100]. Furthermore, low availability of high-quality molecular specimens continues to hamper efforts to build genomic resources for non-avian reptiles, especially when using emerging technologies and approaches capable of constructing highly contiguous genome assemblies (e.g., long read sequencing). Nonetheless, the continued paucity of non-avian reptile genomes may have a more practical source. For example, the small number of non-avian reptile genomes may simply be a consequence of the smaller number of researchers studying reptiles and, ultimately, the smaller sector of society that is engaged in reptile-related activities and community science. Finally, because reptile genomes are on average about twice as big as avian genomes, the sheer cost and labor required to assemble a high-quality reptile genome may be prohibitive. With the advent of increasingly inexpensive long-read sequencing, the production of high-quality reptile genomes may finally ramp up and achieve cruising speed.

## 7. First-Generation Phylogenomic Data Acquisition: Reduced Representation Approaches

Less than a decade after the first draft of the human genome was published [107,177], the melding of two new technological innovations would enable researchers to amass datasets of hundreds to thousands of loci—one to two orders of magnitude more loci than was hitherto possible to acquire using PCR-based approaches [8]. A major advance was the development of massively parallel or “short-read” DNA sequencing, a genomic sequencing platform that far surpassed the data output of the classical Sanger sequencing platform [178,179]. Despite the sudden appearance of several types of short-read genome sequencers during the mid-2000s, only one of them—the Illumina sequencer [180,181]—would come to dominate the genome sequencing and phylogenomics scenes [8], a situation that has remained largely unchanged to the present.

Since this technological breakthrough, phylogenomic studies have routinely utilized genome-wide data consisting of hundreds to thousands of loci (“loci” defined here as DNA segments of at least ~200 base pairs [bp] in length) to estimate species trees and associated historical demographic parameters [8]. A key advantage to using such “big data” in analyses assuming a multispecies coalescent model [182] compared to the one- to several-locus datasets of early molecular phylogenetic and phylogeographic studies is that confidence intervals around parameter estimates are expected to be far more precise (i.e., narrower) than estimates obtained from smaller numbers of loci. This is because the gene tree for each locus is thought to approximate an independent realization of the coalescent process [183,184,185,186]. Accordingly, statistical precision surrounding parameter estimates can be improved simply by increasing the numbers of independent loci [8,187,188,189]. Indeed, several empirical studies have corroborated this basic tenet of multilocus population genetics [190,191,192,193].

Another innovation especially important in phylogenomics was the development of a broad array of molecular techniques for constructing sequencing libraries. Many of these techniques enrich for particular regions of the genome in various ways, which enables a reduced representation of the genome to be preferentially generated for targeted loci [194]. Most early phylogenomics investigations have favored reduced representation approaches for economic reasons, as the cost of sequencing remains a major financial bottleneck for most research groups and sequencing a small percentage of the genome (typically < 5%) results in proportional savings in sequencing cost. The innovation most used for phylogenomics investigations is in-solution hybrid selection—a methodological spinoff of the microarray technology from the early 2000s [195]. In-solution hybrid selection or “target capture” allowed researchers to selectively sequence only targeted DNA sequence loci using Illumina sequencing. By hybridizing 60–120 bp biotinylated RNA oligonucleotide probes to complementary target genomic fragments in the reaction mixture for each sample (individual), DNA sequence data obtained from the probe-annealing and flanking regions could be obtained for hundreds to thousands of genome-wide loci (see reviews in Jennings [8] and Andermann et al. [196]). Therefore, the probe set, often called a “bait set,” can effectively “fish out” the DNA fragments containing the sequences that are complementary to the probes—and, importantly, the accompanying flanking sequences—from a solution of random genomic DNA fragments (i.e., the shotgun library). Once these fragments containing the target sequences have been isolated, they can be sequenced with adequate coverage per locus and analyzed accordingly (Figure 4). Consequently, the standard Illumina sequencing-target capture workflow can regularly churn out immense multilocus datasets in phylogenomics studies in a cost-effective manner.

The first two probe sets for obtaining hundreds to thousands of phylogenomic loci in animals were the “ultraconserved elements loci” (so-called “UCEs” [197]) and “Anchored Hybrid Enrichment loci” (so-called “AHEs” [198]) probe sets (Figure 4). Both probe sets have since propelled studies involving many groups of non-avian reptiles (e.g., [199,200,201]). Although all these probes were designed to hybridize to highly conserved genomic sequences in the genomes of tetrapods and vertebrates, respectively, the template sequences used to design the probes fundamentally differed between UCEs and AHEs. The UCE loci probes anneal to non-coding elements called “ultraconserved elements” or “UCEs,” which have remained virtually unchanged for hundreds of millions of years [202] and may function as regulatory elements that control gene expression of nearby genes. Although their highly conserved nature makes UCEs ideal probe targets in species, these sequences contain insufficient numbers of variable sites to be useful in phylogenomic studies. Accordingly, the less-conserved flanking sequences surrounding the actual UCEs are used in phylogenomic studies (Figure 4). Moreover, because these flanking sequences contain a spectrum of sites ranging from completely non-conserved to highly conserved sites, UCE loci have been informative at both “shallow” (<ca. 10 million years ago [MYA]) and “deep” (>ca. 10 MYA) timescales [197]. The genomic targets of AHE probes, on the other hand, are highly conserved exons whose flanking sequences are useful for shallow and deep timescales [198].

An early example of a reptile-specific, mixed-marker probe set emerged when Singhal et al. [203] designed an all-encompassing probe set specifically for squamate reptiles termed “SqCL set”, which contains probes for harvesting 5052 UCEs, 372 AHEs, and ~50 other “legacy loci” that have been useful for reconstructing the squamate tree of life [204,205]. A survey of the literature over the past five years shows the tremendous impact that UCE and AHE loci have had on phylogenomic studies of non-avian reptiles, as 25 of the 35 target capture studies (71%) employed either or both probe sets (Table 1). Moreover, both sets of loci have been successfully applied to non-avian reptile taxa on both shallow and deep timescales (see [203]; Table 1). Consequently, in one sense the UCE and AHE probe sets are analogous to the first “universal” PCR primers [206], which, together with PCR, launched the field of molecular phylogenetics [207] and modernized phylogeography [208].

Although target capture has been the most commonly used approach in non-avian reptile phylogenomics, other reduced representation methodologies have been applied as well. In an approach commonly referred to as RAD-seq (restriction-site association DNA sequencing; also commonly called genotyping by sequencing [GBS]), one [247,248,249] or more [250] restriction enzymes are used to enrich homologous regions of the genome in a flexible and economical way (Figure 4). The RAD-seq approaches have been commonly applied for population genomics investigations and are occasionally used in phylogenomics studies, although only at shallower phylogenetic scales in three studies of non-avian reptiles (Table 1). Known issues with stochastic locus fallout due to accumulating variation in restriction sites restrict RAD-seq investigations to relatively shallow phylogenetic scales [251,252]. In theory, transcriptomics could be used to generate homologous sequencing data from protein-coding regions of the genome via RNA-seq or similar techniques (Figure 4) [253,254]. However, except for the purposes of constructing probes for sequence capture studies, transcriptomics approaches have not been widely applied in non-avian reptiles due to the difficulty in working with RNA and the small number of samples with sufficient quality available for RNA-seq investigations.

The flexibility of target capture allows researchers to mix target loci from different previously constructed capture panels and even include new loci of interest. Indeed, in many of the studies listed in Table 1, researchers opted to develop their own probe sets for obtaining thousands of annotated exon sequences. For example, many of these studies (e.g., [149,223,224]) used a de novo transcriptome approach to develop a custom exon-capture probe set for each study species group (see [254,255] for the protocol). Although this do-it-yourself approach to designing a custom exon-capture probe set increases project costs in terms of laboratory and bioinformatics work and in consumables, this approach enjoys several advantages over the universal bait kits described earlier for shallow-scale phylogenomic studies: (1) a reference genome is not needed to design loci probe sets; (2) multiple sequence alignments are simpler for coding (e.g., exons) than non-coding (e.g., UCE loci) DNA sequences; (3) larger numbers (>1000) of exonic loci can be developed using the de novo transcriptomic approach compared to the 400–500 loci found obtainable from an AHE set; and (4) thousands of exonic loci that exhibit a wide range of evolutionary rates can be harvested, which contrasts with UCEs and AHE loci [254,255]. However, for studies at deep timescales, universal probe kits will likely perform best and enable researchers to cost-effectively outsource library construction and sequencing to service providers. The advent of massively parallel, reduced representation approaches, especially target capture, have unquestionably revolutionized phylogenomics, as researchers have been able to affordably infer species trees and associated historical demographic parameters with unprecedented accuracy and precision.

## 8. Genome-Scale Phylogenomics: In Silico Investigation of Markers Extracted from Whole Genomes

In a landmark study, Jarvis et al. [256] inferred the higher-level relationships in the avian tree of life using newly generated complete genome sequences for 48 species, ushering in an era of truly genome-scale phylogenomics. Since then, there have been continual advances in short-read sequencing in terms of sequence output and reduction in cost per Gigabase (Gb) of sequences, as well as the development of high-quality, long-read DNA sequencing (e.g., PacBio platform). These improvements in genomic sequencing are now making it practical for researchers working on non-avian reptiles to not only acquire a chromosome-level reference genome assembly for their study organism, but also to obtain large numbers of resequenced genomes for that species or for species in the clade of interest. Indeed, early examples of this approach are already occurring: the California Conservation Genomics Project (CCGP), which is a consortium of 114 principal investigators, is nearly finished with the amassing of high-quality reference genomes and 100–150 resequenced genomes for each of 235 focal species found in marine and terrestrial habitats throughout the state of California [257]. Eight of these species are non-avian reptiles and thus a total of eight high quality reference genomes—three of which were recently published [258,259,260]—plus associated resequenced genome sequences (total of ~800–1200 datasets) will soon be completed. It therefore appears certain that the numbers of population genomic and phylogenomic studies based solely on full genome sequences will accelerate in the future. However, despite these anticipated developments, reduced representation approaches will continue to play an important role in phylogenomics because of cost effectiveness and the wide availability of genetic specimens in natural history museums [242,255] or elsewhere that are degraded or otherwise not suitable for building high-quality genomic resources.

As is the case with reduced representation approaches, the quality of genomics data can have a large impact on the ability of researchers to perform downstream phylogenomic investigations and must be taken into account. The quality of genome assemblies, the forthcoming foundation for truly phylogenomics-scale research, can also be quite variable due to differences in the genomic characteristics of organisms and the practices used to generate genomics datasets and digitally assemble a representation of the genome of an organism, which have changed significantly over time. Moreover, although there are a growing number of high-quality genomes available for amniotes, a “complete genome” is difficult to construct and carries a high burden of proof that has rarely been met, although recently, a first complete, “telomere-to-telomere” reference genome was constructed for human [261]. Equally important for most downstream biological investigations that use genomic resources is a high-quality annotation of all repetitive elements, protein-coding genes, and other important features of the genome that can form the foundation for phylogenomics studies and contribute to genome biology and evolution. Annotation quality is a function of the underlying genome assembly quality, the quantity and quality of functional genomics data used as biological evidence to guide annotation (especially RNA-seq data, but other types of omics data could also be used), and the bioinformatic approach used for annotation. Moreover, additional steps are necessary to estimate homology of genomic loci, such as annotated protein-coding regions, across genomes, a critical prerequisite for phylogenomics studies. Therefore, when evaluating publicly available reference genomes for use in a phylogenomics investigation, it is important to evaluate the quality of genomic resources and build other quality control considerations into the analysis of these comparative genomics datasets.

One major advantage that the complete genome approach enjoys over reduced representation approaches is that computational, or in silico, acquisition of hundreds to thousands of DNA sequence loci from complete genome sequences is much simpler than the target capture workflow [193,262]. For example, to illustrate the phylogenomic utility of UCE markers, McCormack et al. [263] designed a set of in silico UCE probes using available genome data and then performed in silico target capture of target UCE loci from 29 genome sequences for placental mammals. Although there are simpler in silico-based methods for acquiring a comparable dataset from complete genome sequences (as acknowledged by McCormack et al. [263]), their study nonetheless hinted at the promise of in silico extraction of phylogenomic loci from whole genome sequences. Moreover, Costa et al. [193] later designed a Python-based software pipeline that can, in automatic fashion, extract the target loci sequences from complete genome sequences, perform multiple sequence alignments of each locus, and output ready-to-analyze data files. A test run of this program using the human, chimpanzee, gorilla, and orangutan genomes quickly produced a 242 AHE locus dataset. Although analyses of far larger numbers of complete genome sequences would require more time for the software to finish the analysis, the time needed to generate a phylogenomic dataset will undoubtedly still be less than the one- to several-week time requirement for the target capture workflow.

Perhaps an even more important advantage of the complete genome approach is that it will provide researchers, for the first time, an effective way to obtain orthologous sequences from multiple individuals’ genomes for hundreds to thousands of “anonymous loci” [193]. Anonymous loci, which comprise a distinctive marker class first developed by Karl and Avise [264], are ideal DNA sequence markers for phylogeographic and shallow-scale phylogenomic analyses that employ the multi-species coalescent because of their neutral or near-neutral characteristics [8,193,265]—UCEs, AHEs, and other exonic loci violate the neutrality assumption to some degree, making their application to these types of studies uncertain (see [8]). Historically, anonymous loci datasets have been notoriously difficult to obtain, as genomic cloning methods and allele separation methods such as single-stranded conformation polymorphism gels or PCR cloning were the only means by which these types of data could be acquired [190,191,264,266]. Even target capture has done little to help increase the use of anonymous loci in phylogenomic studies because a reference genome is required to generate template sequences for probe kit design—an expensive process that must be iterated for every study because probes for one species or organismal group will likely not perform well for another given the lack of sequence conservation in these markers and their flanking sequences. In silico-based searches for anonymous loci in complete genome sequences are not impacted by these problems, making it straightforward to extract these sequences, align them locus by locus, and output the data in common file formats ready for phylogenomic analyses (Figure 5). As a proof-of-concept illustration of this approach, software called ALFIE (**A**nonymous **L**oci **Fi**nd**e**r; Figure 5) developed by Costa et al. [193] extracted sequences for 292 presumably neutral and genealogically independent anonymous loci (average locus length ~1 kb; total of 292,169 nucleotide sites) from the human, chimpanzee, gorilla, and orangutan genomes. Given that half of the studies listed in Table 1 focused on clades with shallow-time divergences, in silico acquisition of anonymous loci from complete genomes will likely have a large positive impact on phylogeographic, population genomic, and shallow-scale phylogenomic studies in the future.

## 9. Allele Phasing Is a Much-Neglected Component of Most Phylogenomic Workflows

Both the hybrid-capture and in silico approaches to isolating loci for phylogenomics routinely miss a key component of the phylogenomic workflow: allele phasing (Figure 4). Allele phasing tries to reconstruct the actual alleles that comprise a locus over a region of the genome in a diploid organism. Phased alleles are the most natural way to represent genetic diversity within and among species, yet most if not all phylogenetic trees for non-avian reptiles, whether using coalescent or concatenation approaches, neglect to attempt to resolve the two alleles that comprise all loci of diploid organisms. We will not review the different types of allele phasing here, except to say that the approach became popular in the early 2000s with software such as PHASE and fastPHASE [267,268]. Most phylogenomic studies, knowingly or unknowingly, analyze loci that do not represent natural alleles because they are unphased and are usually arbitrary amalgamations of the two alleles found at a particular locus. Neglecting to phase loci has been a major, unacknowledged gap in the program of molecular systematics ever since DNA began to be used routinely in the 1980s. Several studies have demonstrated convincingly that allele phasing improves phylogenetic and phylogeographic inference at multiple temporal scales [196,269,270]. The phylogenomics community likely misses many intriguing insights due to the rampant lack of phasing. Reptile phylogenomics and phylogenomics generally should work towards making allele phasing a routine part of phylogenomic workflows.

## 10. New Reptile Genomes Will Fuel the Future of Reptile Phylogenomics and Genome-Phenotype Discovery via Comparative Genomics

As we have seen, there are critical differences in the production of UCE data in the wet lab, via hybrid capture, and bioinformatically from whole genomes. For example, hybrid capture approaches may result in loci with few flanking regions, especially if the source DNA is degraded, as it often is with historical museum specimens [271]. By contrast, UCEs harvested in silico from whole genomes provide the flexibility to modulate the length of the flanking regions, allowing the researcher to find a balance between maximizing the number of variable sites in the flanking regions with the uncertainty that comes with the inevitable degradation of the alignments of those regions [272]. Consequently, although in silico methods rely on expensive production and assembly of whole genomes, this approach will likely become the norm in phylogenomics of non-avian reptiles.

Another reason why whole genomes will help drive a new generation of reptile phylogenomic studies is that they immediately make available a wealth of marker types that will allow easier comparison of loci of different evolutionary dynamics and phylogenetic information content. A major question in phylogenomics today is what is the optimal marker for a phylogenomic study? This question, in turn, depends somewhat on the method by which phylogenies will be built; concatenation versus coalescent approaches. Regardless of one’s predilections towards one or the other method, a recent study [273] showed that, across a wide variety of phylogenomic data sets, there was strong evidence in the sequence data for heterogeneity and lack of concordance among gene trees—sufficient evidence for many researchers that coalescent approaches, which attempt to accommodate such heterogeneity, should be favored. Whereas concatenation approaches need not pay much attention to the information content of individual loci, relying instead on the summed signal across loci, coalescent approaches—especially “two-step” approaches that build gene trees from each locus prior to amalgamating their signal in a species tree—depend critically on well-resolved gene trees [243,274]. Several phylogenetic and phylogeographic models based on the multispecies coalescent model rely on so-called “sequence-based markers” [275]—sets of aligned sites from which gene trees can be built [276,277]. Sequence-based markers, of which UCEs are one type, constitute a major data type for modern phylogenomics, and whole-genome sequences will maximize the ability to choose among various marker types judiciously. A wealth of phylogenomic studies have shown that there is a great variety of information content of different marker types: for example, introns routinely surpass exons in phylogenomic performance and display less evidence for clade-wide or lineage-specific shifts in base composition, which can compromise many methods of phylogenetic inference [110,278,279]. However, there are many maker types that have been unexplored to date: for example, we know nothing about the performance of loci occurring between genes—intergenic regions. Such regions are likely to be highly heterogeneous, consisting of transposable elements, non-coding regulatory regions and other types of genomic regions with diverse evolutionary dynamics. Such regions, however, need to be explored, not only to further resolve the tree for reptiles but also to learn about the relative performance of all regions of the genome, rather than the select few that have risen to high popularity in recent years. New ‘pangenome’ approaches, such as the ProgressiveCactus genome aligner [280] and the Optimized Dynamic Genome/Graph Implementation [281,282] implemented in the Pangenome Graph Builder—methods that eschew a single reference genome and instead align and compare genomes in an ‘all versus all’ manner—are able to retain all regions of a genome of every species in a comparative study, and are therefore better able to capture complex but potentially phylogenetically informative ‘rare genomic changes’ across the tree of life.

Finally, whole genomes will be essential to the nascent field of “PhyloG2P”—using phylogenies to connect genomes and phenotypes across the tree of life [283]. PhyloG2P, also known as PhyloGWAS [284], presents an extremely exciting prospect of mapping genes underlying key phenotypes using comparative genomics. Several papers in recent years have demonstrated the power of comparative genomics for understanding the loci, both coding and noncoding, that appear to drive specific phenotypes in specific lineages [285,286,287,288]. Examples from amniotes and other taxa reveal PhyloG2P to be a viable endeavor to understand the genetic basis of convergent and lineage-specific traits, such as loss of flight in birds [289], longevity in mammals and fish [290,291,292,293], and limb and digital morphology in mammals and squamates [287,294], and several other traits. Additionally, there is an emerging set of statistical models that allows researchers to study evolutionary associations between candidate regions of the genome and the evolution of specific traits on phylogenies [295,296,297,298]. However, to our knowledge, these promising approaches have rarely been attempted in non-avian reptile datasets [294]. This shortcoming is evident despite the unique phenotypes in this clade, including the many novel genomic features reviewed here, diverse modes of reproduction and sex determination (Figure 1), and numerous, often derived morphological and physiological traits with poorly known genomic underpinnings, such as ectothermy, venom, and limb reduction or loss [32,51,53,54,118,299,300].

## 11. Conclusions

In conclusion, we have reviewed the many novel features of non-avian reptile genomes and the challenges they present for genome assembly, phylogenetic inference, and comparative biology. The relatively large genomes of non-avian reptiles, their sometimes high-density of repetitive elements, and the dearth of researchers straddling the connections between genomic and phenotypic evolution have slowed progress in whole genome sequencing and phylogenomics in non-avian reptiles. Indeed, debate remains about the phylogenetic relationships among squamate reptiles—the most species-rich group of non-avian reptiles. Nevertheless, the wealth of distinctive features of non-avian reptile genomes and phenotypes makes them a prime focus for comparative genomics and phylogenetics. Whole genome sequencing not only provides a rich resource for in silico harvesting of information-rich markers for phylogenomics, but also can provide a platform for finding connections between genomes and phenotypic evolution. We look forward to a new era of integration of non-avian reptile comparative biology, natural history, and genomics, fueled by an increased number of high-quality genomes.

## Figures and Tables

**Figure 1 animals-13-00471-f001:**
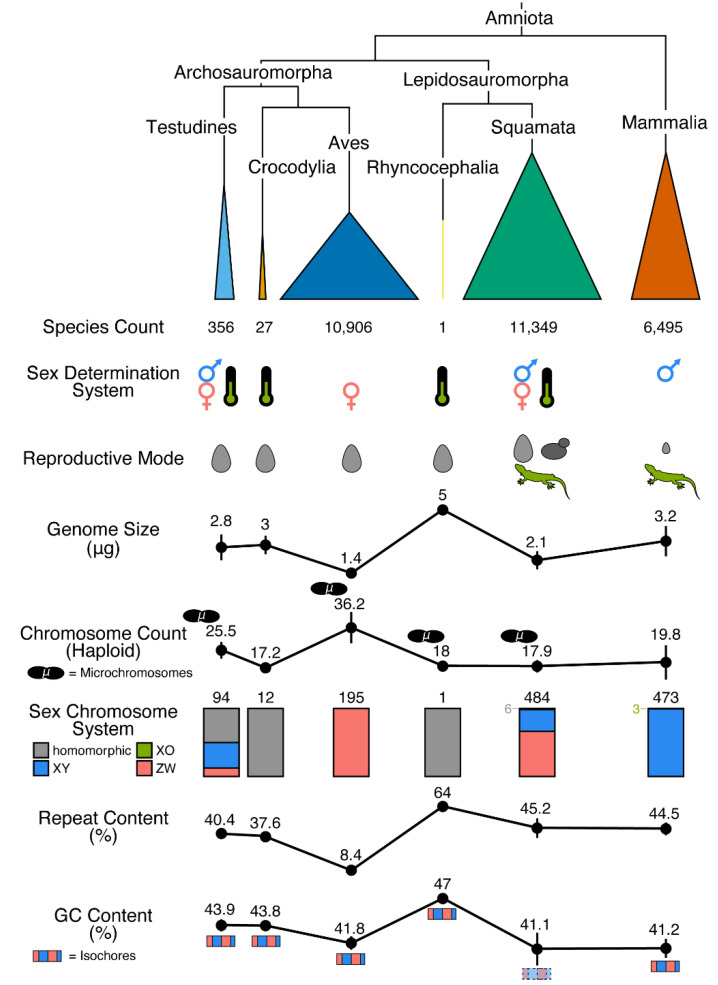
Overview of the natural history of amniotes, including non-avian reptiles, in a phylogenetic context. The width of clades on the phylogeny is proportional to species diversity, which are noted for each clade. For sex determination, GSD is denoted by the male and female symbols for male and female heterogamety, respectively, and TSD is denoted by the thermometer symbol [8,9]. Reproductive mode is indicated with an egg (oviparity), a lizard (viviparity), and a budding yeast symbol (parthenogenesis) [8,10,11,12]. Note the small egg for mammals that reflects the oviparous Monotremata (5 extant species). For genome size (C-value), data from the Animal Genome Size Database [13] were averaged per species and the clade-wise average was calculated as the mean of these species estimates. Karyotype is reported as the mean number of haploid chromosome counts per clade based on the ACC database (https://cromanpa94.github.io/ACC/ (accessed on 1 December 2022)) and lineages with microchromosomes present are indicated with a symbol near the mean chromosome count. Sex chromosome data were gathered from the Tree of Sex database [14]: the proportions of homomorphic, XY, XO, and ZW sex chromosome systems for each clade are indicated with the total species sample size per clade. The small number of squamates with homomorphic sex chromosomes (N = 6) and mammals with XO sex chromosomes (N = 3) are noted, and for counting purposes, complex XY and ZW systems were set to XY and ZW systems, respectively. For repeat content (reported as percentage of the total genome), data from the literature (see [15,16,17,18,19,20,21] and references therein) were averaged per clade. For GC content (reported as percentage of the total genome), data retrieved from the NCBI Genome Assembly database [22] were averaged per species and the clade-wise average was calculated as the mean of these species estimates. Clades with isochore structure are indicated with symbols below the GC estimate [23,24,25,26,27,28,29,30,31] and the isochore symbol for Squamata has a broken border and faded color to indicate the partial loss of isochores in some proportion of species in that lineage. Bars behind the data points are standard deviation. Data gathered from databases were retrieved on 1 December 2022. This figure was inspired by Janes et al. [32].

**Figure 2 animals-13-00471-f002:**
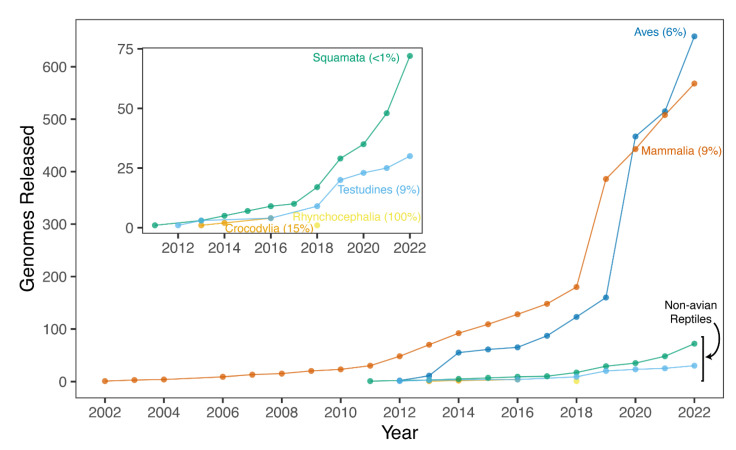
Temporal accumulation of genomes available on NCBI for major amniote clades (data retrieved 1 December 2022). Inset: Details of the growth in the number of available genomes for non-avian reptiles. Note: The counts from this dataset represent a subset of the full non-avian reptile genomes dataset presented in Figure 3, as many genomes are available from sources other than NCBI. This figure was inspired by Bravo et al. [148].

**Figure 3 animals-13-00471-f003:**
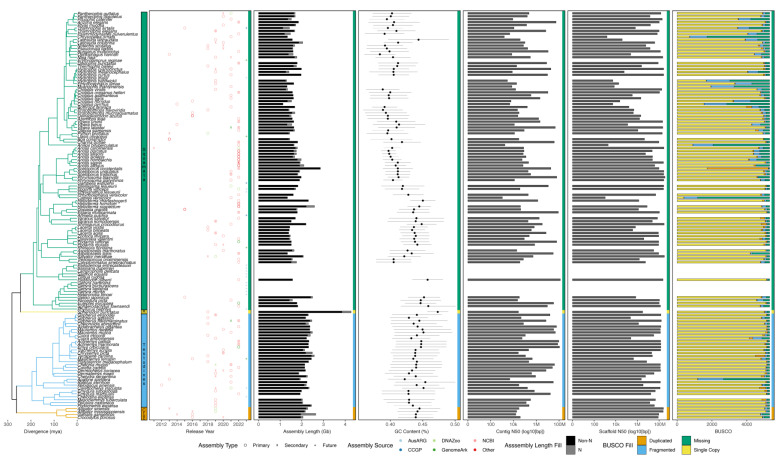
Phylogenetic summary of available reference genomes for non-avian reptiles. The topology and divergence times were gathered from the TimeTree database (accessed 1 December 2022) [4,5]. For taxa that were not already included in TimeTree, we used existing studies of *Gehyra* [149,150,151,152], *Heloderma* [153], *Physignathus* [154], *Gopherus* [155,156], *Actinemys* [157], *Cuora* [158,159], and *Myanophis* [160,161] to place taxa and determine the approximate divergence time. Horizontal bars delineate the major clades: Squamata, Rhyncocephalia (“R”), Testudines, and Crocodylia (“Croc”). The colored bars to the right of each panel indicate each clade and aid in visualization. Publicly-available and announced genomes were collated from NCBI, the Genome10K/VGP/EBGP GenomeArk website (https://genomeark.github.io/genomeark-all/ (accessed on 1 December 2022)), the DNAZoo website (https://www.dnazoo.org/assemblies (accessed on 1 December 2022)), the Australian Amphibian and Reptile Genomics (AusARG) initiative website (https://ausargenomics.com/ (accessed on 1 December 2022)), the California Conservation Genomics Project (CCGP) website (https://www.ccgproject.org/reptiles (accessed on 1 December 2022)), and other locations noted in the literature. For each assembly, we gathered the release date, total assembly length and number of ambiguous (N) bases, and calculated scaffold N50 and contig N50 after breaking scaffolds at runs of >25 Ns. We also ran BUSCO v. 5.4.2 [162] in ‘genome’ mode with the tetrapoda_odb10 dataset to assess the completeness of genomes based on 5310 generally conserved, single-copy tetrapod genes and used bedtools v. 2.29.0 [163] and seqtk v. 1.3-r106 (https://github.com/lh3/seqtk (accessed on 1 December 2022)) to calculate GC content in 500 kb genomic windows (where a minimum of 250 kb of non-N bases were present). Some genomes were not contiguous enough for GC content distributions to be estimated. Where multiple assemblies were available for a species, we plotted the release date and source of each assembly but only quantify genomic characteristic and quality metrics for the primary assembly with the highest-quality assembly based on contiguity and BUSCO results, most of which were designated as the primary assembly on NCBI. Secondary assemblies are those additional assemblies for a given species and future assemblies reflect forthcoming genomes for species that were publicly announced where data are not yet available.

**Figure 4 animals-13-00471-f004:**
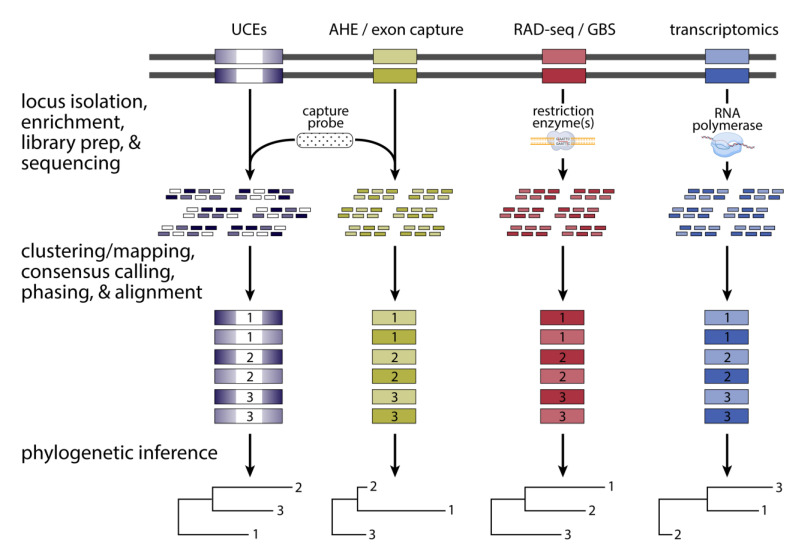
Graphical overview of various reduced representation approaches used in phylogenomics investigations. Alternative depictions are presented for different methods of enriching for particular loci in the genome: two kinds of target capture (targeting UCEs and AHEs or exons), RAD-seq (also known as GBS), and transcriptomics. In each case, the color indicates the location of phylogenetically informative signal in the locus, which typically comprises the whole extent of the target locus, except in the case of UCEs, where this signal is found in the regions flanking the locus. These classes of loci, or markers, are depicted along a diploid genome for a single sample, with heterozygous variation in the form of two alleles at each locus indicated with alternative shading. Although only a single sample is indicated, these approaches would be applied to all samples of interest in parallel, ultimately resulting in sequencing for all samples (e.g., N = 3 samples depicted below). For target capture, the genome is fragmented, and oligonucleotide probes are used to enrich for the target loci. For RAD-seq and transcriptomics, regions of interest are isolated and enriched simultaneously by restriction enzymes and cellular RNA polymerase transcription activity followed by in vitro reverse transcription, respectively. Importantly, of the three general methods, only target capture requires a priori sequencing data and knowledge to construct oligonucleotide probes. After this isolation and enrichment step, all methods proceed generally the same way with standard library preparation and sequencing steps. The resulting sequencing data are also generally analyzed similarly by bioinformatically parsing data to recover sample-specific sequences (three samples are indicated) and clustering sequences by similarity to enable consensus calling (not shown), although a reference genome can aid in this process. Variation across loci is ideally phased to recover the original heterozygous state—two phased alleles per sample are depicted. Phased sequence data for each sample and locus can then be aligned and used for phylogenetic inference.

**Figure 5 animals-13-00471-f005:**
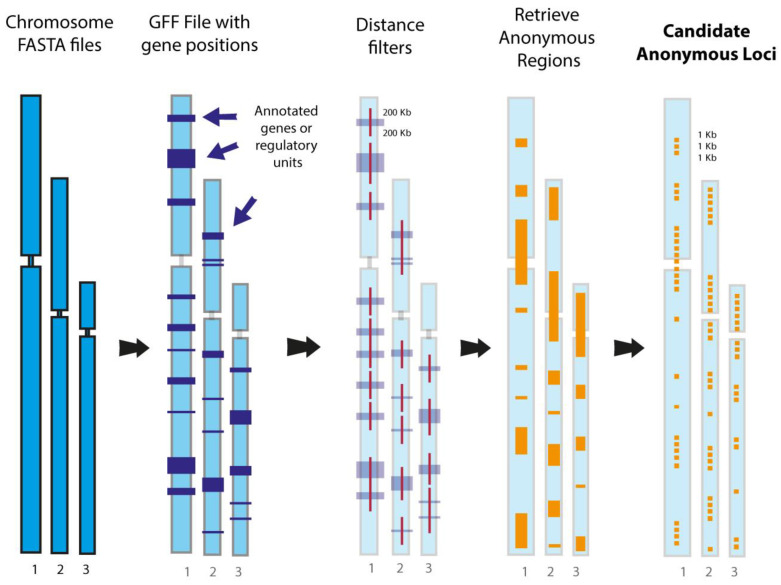
The ALFIE software pipeline for in silico extraction of anonymous loci sequences from complete genome sequences and assembling ready-to-analyze data sets. The user first inputs genome sequences in FASTA format, one of which must be a reference genome with a GFF (general features format) file of genomic annotations, namely protein-coding genes, and regulatory regions. The program then maps the presumably neutral intergenic or “anonymous” regions by applying a user-specified physical distance threshold (in base pairs [bp]). This filter discards all chromosomal regions that contain known functional elements and their flanking sequences (up to the threshold distance), thereby helping to ensure that retained anonymous regions are unaffected by natural selection (e.g., background selection). The anonymous regions are then split into user-specific locus lengths (in bp), which are referred to as “candidate anonymous loci.” In the final steps (not shown), the program uses candidate anonymous loci as query sequences to conduct BLAST searches against all input genomes, keeping only single-copy loci in all genomes, before saving them to a FASTA file. Next, the program conducts multiple sequence alignments for all loci before using a second user-defined distance threshold (in bp) to retain loci that are spaced far enough from other sampled loci that they likely meet the independent gene tree assumption. Lastly, the program outputs the dataset in NEXUS, PHYLIP, and FASTA formats, and can use other included modules to find in automated fashion the best DNA substitution model and gene tree for each locus (figure modified after Figure 1 in Costa et al. [193]). See also Jennings [189] for further explanation and extensions of physical distance threshold theory. Reprinted with permission from Costa et al. [193].

**Table 1 animals-13-00471-t001:** Phylogenomic studies of non-avian reptile clades published between 2017 and 2022 that used at least 100 DNA sequence loci.

Study	Taxon	Type of Loci	# of Loci	# Samples	Depth of Divergences
Ashman et al., 2018 [209]	lizards	exons	547	64	shallow
Blair et al., 2022 [210]	lizards	UCEs	3157	34	shallow
Blom et al., 2017 [211]	lizards	exons	2840	28	shallow
Blom et al., 2019 [212]	lizards	exons	2457	135	shallow
Bragg et al., 2018 [213]	lizards	exons	2364	123	shallow-deep
Brennan et al., 2021 [214]	lizards	AHEs	388	103	shallow-deep
Bryson et al., 2017 [215]	lizards	UCEs	3282	58	shallow
Domingos et al., 2017 [216]	lizards	AHEs	422	30	shallow
Freitas et al., 2022 [217]	lizards	exons	625	69	shallow
Garcia-Porta et al., 2019 [218]	lizards	AHEs + other	6593 (324 AHEs + 6269 other)	262	shallow-deep
Grummer et al., 2018 [219]	lizards	UCEs + exons	589 (541 UCEs + 44 exons)	29	shallow
Morando et al., 2020 [220]	lizards	UCEs + exons	588 (540 UCEs + 44 exons)	26	deep
Moritz et al., 2018 [149]	lizards	exons	1636	56	shallow
Panzera et al., 2017 [221]	lizards	UCEs	581 (538 UCEs + 43 exons)	16	higher
Ramírez-Reyes et al., 2020 [222]	lizards	RAD-seq	78,970–549,193	90	shallow
Reilly et al., 2022a [223]	lizards	exons	709	99	shallow-deep
Reilly et al., 2022b [224]	lizards	exons	1154	104	shallow
Reynolds et al., 2022 [225]	lizards	UCEs	4055	82	shallow-deep
Rodriguez et al., 2018 [226]	lizards	UCEs	2690	119	shallow-deep
Schools et al., 2022 [227]	lizards	UCEs	5060	30	higher
Singhal et al., 2018 [228]	lizards	exons	2668	25	shallow
Skipwith et al., 2019 [229]	lizards	UCEs	4268	290	shallow-deep
Tucker et al., 2017 [230]	lizards	AHEs	316	16	shallow
Wood et al., 2020 [231]	lizards	UCEs	772–4715	42	deep
Zozaya et al., 2022 [232]	lizards	exons	1429	33	shallow
Bernstein and Ruane 2022 [233]	snakes	AHEs + UCEs + other	1–642 UCEs, 1–39 AHEs, 2–11 other	156	shallow-deep
Blair et al., 2019 [234]	snakes	UCEs	3384	54	shallow
Chen et al., 2017 [235]	snakes	AHEs	304	88	shallow
Esquerré et al., 2020 [236]	snakes	AHEs	376	50	deep
Hallas et al., 2022 [237]	snakes	RAD-seq	22,289–48,867	49	shallow
Li et al., 2022 [238]	snakes	exons + other	3023 (1948 exons + 1948 other)	24	deep
Myers et al., 2022 [239]	lizards	RAD-seq	2950	74	shallow
Natusch et al., 2021 [240]	snakes	AHEs	421	19	shallow
Nikolakis et al., 2022 [241]	snakes	UCEs	3383–4146	43	shallow
Ruane and Austin 2017 [242]	snakes	UCEs	2318	10	deep
Burbrink et al., 2020 [243]	squamates	AHEs	394	289	deep
Singhal et al., 2021 [244]	squamates	AHEs + UCEs + other	5462 (372 AHEs + 5052 UCEs + 38 other)	92	deep
Streicher and Wiens 2017 [245]	squamates	UCEs	2738	24	deep
Shaffer et al., 2017 [246]	turtles	UCEs + other	539	24	deep

## Data Availability

The data presented in this study (Figure 1, Figure 2 and Figure 3) are openly available at GitHub/Zenodo at https://doi.org/10.5281/zenodo.7517351.

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
