# Peer review of "Genome Evolution and the Future of Phylogenomics of Non-Avian Reptiles"

_animals, 2023, doi:10.3390/ani13030471_

Round 1

Reviewer 1 Report

In their manuscript, “Genome evolution and the future of phylogenomics of non-avian reptiles”, Card and colleagues set out to make the case for the sequencing of genome scale data for non-avian reptiles due to their comparative diversity among amniotes and their concomitant lack of genome scale data available. The title links genome evolution to the future of phylogenomics in the group. Throughout the paper, the authors make a compelling case for genome scale data to be made available for non-avian reptiles, going into exhaustive detail in terms of both the evolutionary developmental characteristics that make non-avian reptiles unique and in how these characteristics might manifest as signatures on the genome. The manuscript is very clearly written and extensively sourced, with nearly 200 citations! I expect this will be a useful and highly cited review for the non-avian reptile research community.

The authors make a rock-solid case that genome-scale data are sorely needed for non-avian reptiles but I feel as though this review could benefit from discussion of why non-avian reptile phylogenomics is worth doing. My suggestion – and given the overall quality of the manuscript the authors should consider this just a suggestion – would be to add a small section that explicitly states what can be gained through improved phylogenomic analyses in non-avian reptiles. Firstly, the authors have already clearly laid out the unique genome biology of these clades with great diversity in traits that are otherwise invariant in other groups. It follows then, that phylogenetic comparative analyses of these traits could be a particularly fruitful endeavor, one that requires robust and well sampled phylogenies. Second, there remains substantial debate among herpetologist about the phylogenetic relationships among squamates. This is a topic that has garnered much public attention over the past two decades particularly because of the discordance between morphological and molecular phylogenies that is evident for many non-avian reptile taxa, most notably squamates. A phylogeny built using whole genome data is needed in this ongoing debate and I think this manuscript is a good place to make that case,.

The scholarship of this article is very impressive and thorough. After a careful read I found only one occasion where (I believe) an important citation was not included. In the discussion of isochore evolution (Line 304) the authors cite Fujita et al (2011) who found an absence of isochores in the Anolis carolinensis genome. While the authors discuss subsequent squamate isochore diversity found thorugh analysis of snake genomes (Castoe et al 2013) a separate paper by Costantini et al in 2016 (https://doi.org/10.1093/gbe/evw056) calls into question the initial observation that anoles lack isochores at all. While I strongly disagree with the tone of Costantini and colleagues in that paper, I think their observation should be mentioned in this section.

Minor, specific comments

Line 25: “…to collaboratively begin…” is a split infinitive. I suggest “to begin collaboratively”

Line 210: “…, although less obviously homomorphic 210 chromosomes are also possible, but have been more difficult to diagnose.” I believe a word is missing here, perhaps should be “…, although detecting less obvious homomorphic chromosomes is also possible, but have been more difficult to diagnose.”

Line 332: “proportions” should be “proportion” 

Line 338-9: “…there are 165 and 23 publicly available and announced (i.e., expected in the future) non-avian reference genomes, respectively…”   I had to read this sentence a few times to parse “respectively” correctly. I suggest removing respectively entirely in favor of “…there are 165 publicly available and 23 announced (i.e., expected in the future) non-avian reference genomes…”   

P12, subsection 6 (line numbers end at page 9): While noted elsewhere, it might be worth mentioning again here that the technologies that have most advanced genome assembly quality also have the strictest requirements for specimen preparation and storage. While studies routinely use decades old ethanol preserved tissues for reduced representation sequencing, sequencing for de novo assembly virtually always requires collection of new material (which is difficult/impossible in some taxa).

Author Response

In their manuscript, “Genome evolution and the future of phylogenomics of non-avian reptiles”, Card and colleagues set out to make the case for the sequencing of genome scale data for non-avian reptiles due to their comparative diversity among amniotes and their concomitant lack of genome scale data available. The title links genome evolution to the future of phylogenomics in the group. Throughout the paper, the authors make a compelling case for genome scale data to be made available for non-avian reptiles, going into exhaustive detail in terms of both the evolutionary developmental characteristics that make non-avian reptiles unique and in how these characteristics might manifest as signatures on the genome. The manuscript is very clearly written and extensively sourced, with nearly 200 citations! I expect this will be a useful and highly cited review for the non-avian reptile research community.

We thank the reviewer for their kind comments and useful feedback and have addressed all critiques using point-by-point responses, below.

The authors make a rock-solid case that genome-scale data are sorely needed for non-avian reptiles but I feel as though this review could benefit from discussion of why non-avian reptile phylogenomics is worth doing. My suggestion – and given the overall quality of the manuscript the authors should consider this just a suggestion – would be to add a small section that explicitly states what can be gained through improved phylogenomic analyses in non-avian reptiles. Firstly, the authors have already clearly laid out the unique genome biology of these clades with great diversity in traits that are otherwise invariant in other groups. It follows then, that phylogenetic comparative analyses of these traits could be a particularly fruitful endeavor, one that requires robust and well sampled phylogenies. Second, there remains substantial debate among herpetologist about the phylogenetic relationships among squamates. This is a topic that has garnered much public attention over the past two decades particularly because of the discordance between morphological and molecular phylogenies that is evident for many non-avian reptile taxa, most notably squamates. A phylogeny built using whole genome data is needed in this ongoing debate and I think this manuscript is a good place to make that case,.

We agree with the reviewer that more explicit discussion was needed of the potential fruits of phylogenomics investigations of non-avian reptiles. We have expanded upon the existing discussion of these important points made in the Conclusion and explicitly mention lingering questions about the squamate phylogeny and the potential for genomic resources to illuminate unique aspects of genome biology and natural history in non-avian reptiles.

The scholarship of this article is very impressive and thorough. After a careful read I found only one occasion where (I believe) an important citation was not included. In the discussion of isochore evolution (Line 304) the authors cite Fujita et al (2011) who found an absence of isochores in the Anolis carolinensis genome. While the authors discuss subsequent squamate isochore diversity found thorugh analysis of snake genomes (Castoe et al 2013) a separate paper by Costantini et al in 2016 (https://doi.org/10.1093/gbe/evw056) calls into question the initial observation that anoles lack isochores at all. While I strongly disagree with the tone of Costantini and colleagues in that paper, I think their observation should be mentioned in this section.

As suggested by the reviewer, we have added a reference to Costantini et al. (2016) in this section of the text.

Minor, specific comments

Line 25: “…to collaboratively begin…” is a split infinitive. I suggest “to begin collaboratively”

We have made the suggested change.

Line 210: “…, although less obviously homomorphic chromosomes are also possible, but have been more difficult to diagnose.” I believe a word is missing here, perhaps should be “…, although detecting less obvious homomorphic chromosomes is also possible, but have been more difficult to diagnose.”

We have revisited this section of text and have modified the text based on reviewer feedback to improve readability.

Line 332: “proportions” should be “proportion”

We have kept “proportions” but removed the “a” beforehand to correct this sentence. We use the plural form since we are referring to the proportions of multiple small reptilian clades.

Line 338-9: “…there are 165 and 23 publicly available and announced (i.e., expected in the future) non-avian reference genomes, respectively…”   I had to read this sentence a few times to parse “respectively” correctly. I suggest removing respectively entirely in favor of “…there are 165 publicly available and 23 announced (i.e., expected in the future) non-avian reference genomes…”

We have made the suggested change.

P12, subsection 6 (line numbers end at page 9): While noted elsewhere, it might be worth mentioning again here that the technologies that have most advanced genome assembly quality also have the strictest requirements for specimen preparation and storage. While studies routinely use decades old ethanol preserved tissues for reduced representation sequencing, sequencing for de novo assembly virtually always requires collection of new material (which is difficult/impossible in some taxa).

We have followed the reviewer’s suggestion and have mentioned specimen quality considerations in this section of text.

Reviewer 2 Report

Major Comments:

-       This is a very important review paper in that it compiles a wealth of information needed to fully understand the current state, and future possibilities, of reptilian phylogenomics. The paper is extremely well-written with no spelling or grammatical errors that I could find. The information presented all appears to be correct. The ideas flow naturally, and the organization of the paper is well thought out.

-      There are some small formatting issues throughout the manuscript that should be dealt with by the journal’s editors and production team, not by the authors, just to make the paper look clean.

-       Overall this paper will become an excellent resource for all reptile genomics and phylogenomics studies, and will likely be used as a genomics/phylogenomics resource for researchers of other taxonomic groups as well.

Minor Comments:

Table 1 - the table goes off the end of the right side of the page; include a margin.

Table 1 - Reilly et al. 2022a had 99 samples, not 89.

Table 1 -  it might be worth including Blom et al. 2019 which used >1000 exons for >100 Cryptoblepharus skinks samples:

Blom MPK, Matzke NJ, Bragg JG, Arida E, Austin CC, Backlin AR, Carretero MA, Fisher RN, Glaw F, Hathaway SA, Iskandar DT, McGuire JA, Karin BR, Reilly SB, Rittmeyer EN, Rocha S, Sanchez M, Stubbs AL, Vences M & Moritz C. 2019. Habitat preference modulates trans-oceanic dispersal in a terrestrial vertebrate. Proceedings of the Royal Society B, 286:20182575.

Author Response

Major Comments:

-       This is a very important review paper in that it compiles a wealth of information needed to fully understand the current state, and future possibilities, of reptilian phylogenomics. The paper is extremely well-written with no spelling or grammatical errors that I could find. The information presented all appears to be correct. The ideas flow naturally, and the organization of the paper is well thought out.

We have reviewed the reviewer feedback and addressed each point below. We thank the reviewer for their helpful feedback!

-      There are some small formatting issues throughout the manuscript that should be dealt with by the journal’s editors and production team, not by the authors, just to make the paper look clean.

We appreciate the reviewer pointing out these issues and we will work with the journal to ensure that formatting is corrected before publication.

-       Overall this paper will become an excellent resource for all reptile genomics and phylogenomics studies, and will likely be used as a genomics/phylogenomics resource for researchers of other taxonomic groups as well.

We agree with the reviewer’s point. Indeed, this was a major motivation for producing this review.

Minor Comments:

Table 1 - the table goes off the end of the right side of the page; include a margin.

We note this formatting issue as well and will work with the journal to address it.

Table 1 - Reilly et al. 2022a had 99 samples, not 89.

We have corrected this count.

Table 1 -  it might be worth including Blom et al. 2019 which used >1000 exons for >100 Cryptoblepharus skinks samples:

Blom MPK, Matzke NJ, Bragg JG, Arida E, Austin CC, Backlin AR, Carretero MA, Fisher RN, Glaw F, Hathaway SA, Iskandar DT, McGuire JA, Karin BR, Reilly SB, Rittmeyer EN, Rocha S, Sanchez M, Stubbs AL, Vences M & Moritz C. 2019. Habitat preference modulates trans-oceanic dispersal in a terrestrial vertebrate. Proceedings of the Royal Society B, 286:20182575.

As suggested, we have added this study to the table.